



# NH-SWE: Northern Hemisphere Snow Water Equivalent dataset based on in-situ snow depth time series

Adrià Fontrodona-Bach[1], Bettina Schaefli[2], Ross Woods[3], Adriaan J Teuling[4], and Joshua R Larsen[1]

[1]School of Geography, Earth and Environmental Sciences, University of Birmingham, United Kingdom
[2]Institute of Geography, GIUB, and Oeschger Centre for Climate Change Research, OCCR, University of Bern, Switzerland
[3]Department of Civil Engineering, University of Bristol, United Kingdom
[4]Hydrology and Quantitative Water Management Group, Wageningen University and Research, The Netherlands

**Correspondence:** Adrià Fontrodona-Bach (adria.fontrodona@gmail.com)

**Abstract.** Ground-based datasets of observed Snow Water Equivalent (SWE) are scarce, while gridded SWE estimates from remote-sensing and climate reanalysis are unable to resolve the high spatial variability of snow on the ground. Long-term ground observations of snow depth, in combination with models that can accurately convert snow depth to SWE, can fill this observational gap. Here, we provide a new SWE dataset (NH-SWE) that encompasses 11,071 stations in the Northern

Hemisphere, and is available at doi.org/10.5281/zenodo.7515603 (Fontrodona-Bach et al., 2023). This new dataset provides daily time series of SWE, varying in length between one and seventy-three years, spanning the period 1950-2022 and covering a wide range of snow climates including many mountainous regions. At each station, observed snow depth was converted to SWE using an established snow-depth-to-SWE conversion model, with excellent model performance using regionalised parameters based on climate variables. The accuracy of the model after parameter regionalisation is comparable to that of the calibrated

model. The key advantages and strengths of the regionalised model presented here are its transferability across climates and the high performance in modelling daily SWE dynamics in terms of peak SWE, total snowmelt and duration of the melt season, as assessed here against a comparison model. This dataset is particularly useful for studies that require accurate time series of SWE dynamics, timing of snowmelt onset, and snowmelt totals and duration. It can e.g. be used for climate change impact analyses, water resources assessment and management, validation of remote sensing of snow, hydrological modelling and snow

data assimilation into climate models.

## 1 Introduction

The modification of the cryosphere is one of the most visible effects of ongoing climate warming (Beniston et al., 2018). In this context, snow cover is particularly important, as it provides an important seasonal hydrologic buffer over high latitudes and high elevations, through storage over the accumulation season and a delayed release of water in the subsequent melt season

(Kuppel et al., 2017). Snow cover also plays an important role in the Earth's climate through snow-albedo feedbacks (Déry and Brown, 2007). Studies showing the impact of global warming on snow are numerous: declining trends in snow depth have been observed over the European Alps (Matiu et al., 2021a), the Pyrenees (López-Moreno et al., 2020), and all of Europe except the coldest climates (Fontrodona Bach et al., 2018); decreasing snow cover duration and extent have been reported over



the Northern Hemisphere (Bormann et al., 2018; Mudryk et al., 2020) and over 78% of global mountain regions (Notarnicola, 2020); and increasing snowmelt in winter has been observed in western North America (Musselman et al., 2021). Regions whose water resources strongly depend on snow water storage are at risk of large declines in spring and summer streamflow, posing a potential future threat to human water used for ∼2 billion people (Mankin et al., 2015). There is therefore an ongoing need for access to wide coverage and reliable snow data and research.

There are various ways to measure snow, which serve different scientific purposes. Snow depth is the thickness of snow accumulated above ground level and can easily be measured manually with a ruler or automatically with an acoustic sensor that measures snow height. The depth of water that is actually stored in a snowpack is referred to as Snow Water Equivalent (SWE) and corresponds to the depth of liquid water that would result from melting the entire snowpack.

Reliable estimates of SWE, rather than snow depth, are thus needed for water resources assessments across scales. However, limitations arise when estimating SWE at regional, continental or global scales. Remote sensing estimates of SWE provided by satellite measurements are in constant development but currently have low accuracy and are limited to shallow (<150 mm) snowpacks (Luojus et al., 2021). Estimates of SWE resulting from reanalysis and land surface modelling rely on snow models forced with meteorological data and are also subject to biases and errors (Brun et al., 2013; Broxton et al., 2016; Muñoz-Sabater et al., 2021). A common limitation of these gridded SWE estimates is that they cannot reproduce the high spatial variability of snow on the ground, especially for mountain regions and complex terrain (Clark et al., 2011; López-Moreno et al., 2013). A review of global gridded datasets of SWE shows up to a 50% variability in peak snow accumulation between different datasets (Mudryk et al., 2015).

Adding to the complexity of obtaining reliable SWE data, large scale estimates of SWE often have limited validation against observed ground data, which is scarce in time and in space. In fact, both manual and remotely sensed measurement techniques exist for SWE, but they are either complex, time consuming, or require specialised equipment (Jonas et al., 2009; Winkler et al., 2021). In contrast to complex SWE measurements, manual and automatic snow depth measurements are more straightforward and therefore more widespread. Many regional snow depth datasets exist or are emerging, with an increasing number of National Hydrological and Meteorological Services making snow depth data publicly available and easily downloadable through site portals and Application Programming Interfaces (API). However, an estimate of snow density is needed together with a snow depth measurement to obtain the snow water equivalent.

There is an increasing number of models that can accurately convert snow depth to SWE using simple empirical regressions (Mizukami and Perica, 2008; Jonas et al., 2009; Sturm et al., 2010; Bormann et al., 2013; Mccreight and Small, 2014; Pistocchi, 2016; Hill et al., 2019; Ntokas et al., 2021). These regression based methods require paired snow depth-SWE ground measurements to calibrate parameters that later on are used to estimate SWE when only snow depth measurements are available. Some of the approaches generalize parameters regionally, based on elevation and on the day of the year (Jonas et al., 2009), or globally based on climatological variables (Sturm et al., 2010; Hill et al., 2019; Szeitz and Moore, 2023). The common limitation of using regression based approaches is that a SWE value on a given time step is estimated independently for each snow depth value, irrespective of snow depth values on preceding time steps. Therefore, conversion of time series of snow depth to SWE leads to an incorrect temporal evolution of SWE because the regression cannot account neither for the settling of new snow



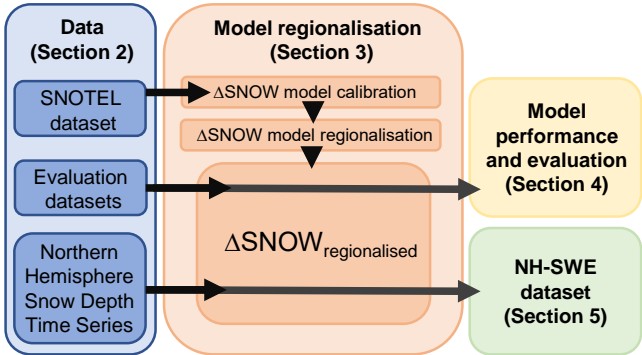

**Figure 1.** Workflow chart to obtain the NH-SWE dataset. The ΔSNOW model is from Winkler et al. (2021).

and nor for the compaction of the snowpack (Jonas et al., 2009). This problem has been addressed by considering the temporal
evolution of snow depth in the conversion model (Mccreight and Small, 2014; Winkler et al., 2021), but, to date, no approach
has generalised this method for regional or global use.

Here we bring together the increasing number of available in-situ snow depth datasets and the increasing number of snow
depth to SWE conversion models to produce a new, freely available SWE dataset: the NH-SWE dataset. This compilation
and analysis provides i) the first pan-Northern Hemisphere in-situ snow depth time series compilation and ii) a conversion
from depth to SWE using an established model (ΔSNOW, Winkler et al. (2021)). This SWE dataset can be extremely useful
across a wide variety of applications such as validation of remote sensing products, hydrological and environmental modelling,
assimilating snow data into models, and general climate research.

This paper is organized as follows: 1) data sources used to calibrate, regionalise and evaluate the model for snow depth to
SWE conversion (Section 2), 2) model development and implementation (Section 3), 3) performance of the model for across a
variety of variables (Section 4), 4) demonstration of some key features of the NH-SWE dataset (Section 5), including its usage
and limitations (Section 6).

## 2   Data sources

The data used to compile NH-SWE from in-situ snow depth observations can be divided in two main groups: group HS-SWE
includes all stations that have both snow depth (Height of Snow, HS) and SWE data available; group HS includes all stations
that have only HS observations available.

Group HS-SWE data can be further split for the purposes of model implementation into a) data used for model regionalisation
and b) data used for independent evaluation. The HS data group then provides the input data to generate the new SWE estimates
within the NH-SWE dataset (Section 5) but is not used in model implementation. A summary chart of the workflow to produce
the final SWE dataset is shown in Fig. 1. An overview of all the data used to derive the NH-SWE dataset is shown in Table 1,
and the spatial distribution of the data and sources is shown in Fig. 2.



**Table 1.** Overview of datasets and sources used for model regionalisation, model evaluation and input for the final NH-SWE dataset. *Model regionalisation* and *model evaluation data* refer to Sections 2.1 and 2.2, and *input data* refers to Section 2.3. $SWE_b$ for daily measurements, $SWE_b$ for biweekly measurements. The number of sites is after quality control and selection. The minimum length is 1 year for all datasets.

| Data Source | Data Type | Data Use | Number of Sites | Length (years) mean | max | Reference |
|---|---|---|---|---|---|---|
| SNOTEL | $HS\text{-}SWE_d$ | Model regionalisation | 812 | **14** | *25* | USDA NRCS (2022) |
| CanSWE | $HS\text{-}SWE_d$ | Model evaluation | 68 | **8** | *28* | Vionnet et al. (2021, 2022) |
| GCOS-CH | $HS\text{-}SWE_b$ | Model evaluation | 8 | **57** | *70* | Marty (2020) |
| RIHMI-WDC | $HS\text{-}SWE_b$ | Model evaluation | 2 | **30** | *34* | RIHMI-WDC (2022) |
| NVE | $HS\text{-}SWE_d$ | Model evaluation | 6 | **3** | *9* | NVE (2022) |
| Kuhtai-AT* | $HS\text{-}SWE_d$ | Model evaluation | 1 | **23** | *23* | Krajči et al. (2017) |
| Sodankyla-FI* | $HS\text{-}SWE_d$ | Model evaluation | 1 | **6** | *6* | Essery et al. (2016) |
| Col-de-Porte-FR* | $HS\text{-}SWE_b$ | Model evaluation | 1 | **18** | *18* | Morin et al. (2012) |
| Alptal-CH* | $HS\text{-}SWE_b$ | Model evaluation | 1 | **1** | *1* | Stähli (2018) |
| ECA&D | HS | Input data for NH-SWE | 3,050 | **40** | *70* | Klein Tank et al. (2002) |
| FMI | HS | Input data for NH-SWE | 204 | **36** | *63* | FMI (2022) |
| Matiu20 | HS | Input data for NH-SWE | 535 | **34** | *60* | Matiu et al. (2021b, a) |
| RIHMI-WDC | HS | Input data for NH-SWE | 543 | **61** | *69* | RIHMI-WDC (2022) |
| GHCNd | HS | Input data for NH-SWE | 6,478 | **23** | *71* | Menne et al. (2012) |
| MeteoSwiss | HS | Input data for NH-SWE | 261 | **33** | *70* | MeteoSwiss (2022) |

*Referred as "single sites" in Fig. 2a

## 2.1 HS-SWE data for model regionalisation

The SNOwpack TELemetry Network (SNOTEL) dataset contains a large network of automated sub-daily observations of HS and SWE data over the western United States and is freely available (USDA NRCS, 2022). The dataset covers a wide range of snow climates and characteristics such as the maritime Cascades mountain range, the continental Rockies and high latitude tundra/taiga (Serreze et al., 1999). The SNOTEL dataset only contains measurement stations where a minimum of 40 days of continuous snow cover is observed on average (since it has been designed to monitor seasonal snowpacks). We use this dataset to calibrate and regionalise the ΔSNOW model (Winkler et al., 2021) over a range of climates. We apply the same data preprocessing to the SNOTEL dataset as Hill et al. (2019), retaining only measurement stations with joint HS and SWE records, and removing outliers. The resulting dataset for model regionalisation contains 812 sites with daily HS-SWE time series data ranging from 1 to 25 years in length (see Table 1 and Fig. 2a).

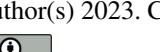

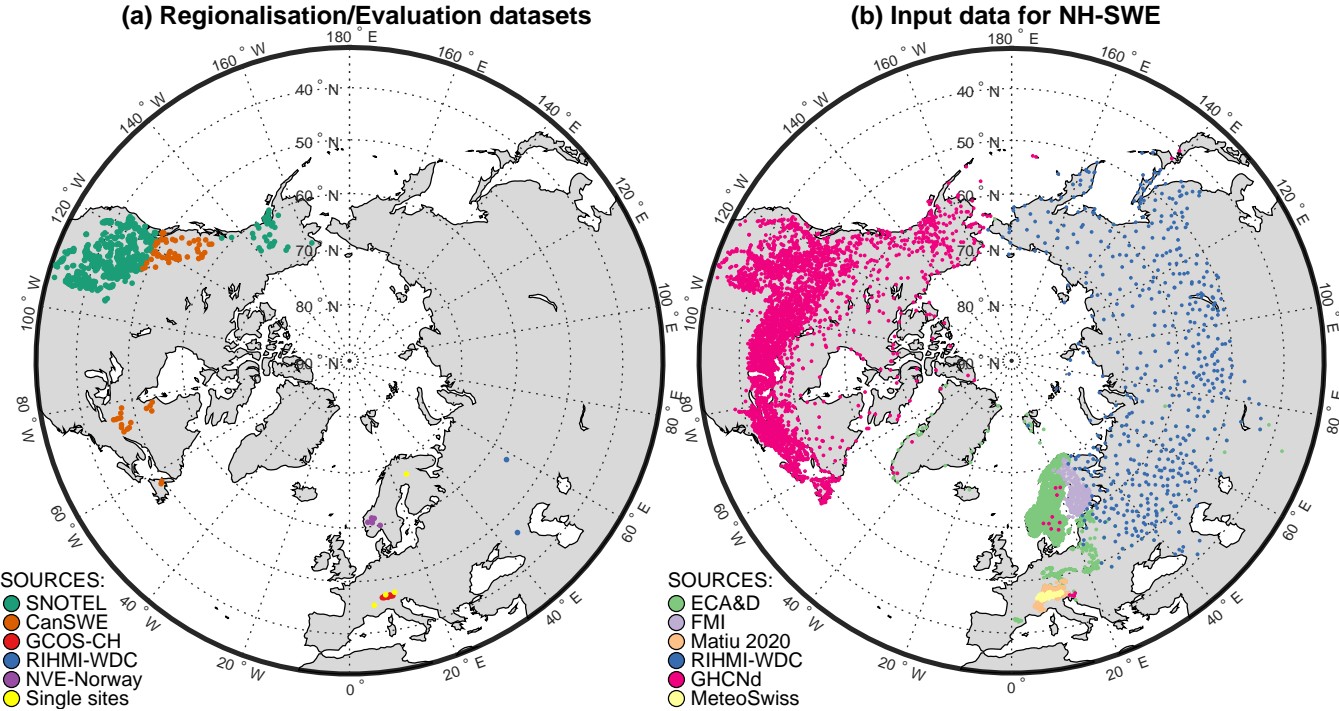

**Figure 2.** Spatial distribution of stations in the datasets used. (a) Paired HS-SWE datasets for model regionalisation and evaluation and (b) HS data used as model input for the final modelled SWE dataset (NH-SWE). In (a), SNOTEL is used for model regionalisation, and the rest are evaluation datasets. See Table 1 for more information on the sources.

## 2.2 HS-SWE data for model evaluation

We independently evaluate the ΔSNOW model using a SWE dataset from Canada (Vionnet et al., 2021), which contains 2,607 stations with historical HS and SWE measurements. Selecting only those stations with daily observations of HS and SWE, and applying gap filling and quality control (see Appendix A), 68 stations distributed over Eastern and Western Canada are retained

for model evaluation (see Fig. 2a and Table 1).

To assess the transferability of the model to outside North America, we compiled 20 additional HS and SWE datasets from 7 different sources: 8 stations from the Global Climate Observing System (GCOS) in Switzerland (Marty, 2020); 2 stations from the All-Russia Research Institute of Hydrometeorological Information - World Data Center (RIHMI-WDC, 2022); 6 stations from the Norwegian Water Resources and Energy Directorate (NVE, 2022); and 4 single station observations from

different sources over the European Alps and Finland (Stähli, 2018; Krajči et al., 2017; Essery et al., 2016; Morin et al., 2012). The number of additional stations available for independent validation outside North America is low because 1) continuous observation of both HS and SWE is much rarer and 2) if collected, is also rarely provided within open data repositories. Most of this additional validation data contain daily observations of SWE, but some contain only weekly or biweekly measurements





(see Table 1). This change in temporal resolution in turn reduces the metrics that can be evaluated. For example, the timing of
snowmelt onset cannot be accurately determined without daily SWE measurements (see Section 3.6).

## 2.3 HS data as model input for NH-SWE dataset

We have gathered and compiled pan Northern Hemisphere datasets of in-situ daily HS observations from different sources.
This relies on the European Climate Assessment and Dataset (ECA&D), which has already been described (Klein Tank et al.,
2002) and analysed in previous studies (Fontrodona Bach et al., 2018), but with some important additions and updates. Gaps
in time series from Finland in the ECA&D have been filled where possible by downloading HS time series directly from the
Finnish Meteorological Insitute (FMI). The under-representation of alpine sites in the ECA&D has been reduced through data
obtained from MeteoSwiss and from data published by Matiu et al. (2021a, b). The ECA&D coverage over Russia has been
completely replaced by data from the All-Russia Research Institute of Hydrometeorological Information - World Data Center
(RIHMI-WDC), which contains longer and updated coverage of many sites. The western half of the Northern Hemisphere is
well covered by data available from the Global Historical Climatology Network daily (GHCNd). In the case of multiple time
series from the same location being available across the compiled datasets, we selected the time series with the most updated
record or the one with fewer gaps. Because of the sharp increase in data availability and quality after 1950 (Fontrodona Bach
et al., 2018; Matiu et al., 2021a) our data compilation begins from 1 September 1949 to 31 August 2022. Most snow depth
measurements are manual with a precision of $\pm$ 1 cm, but automatic measurements with higher precision are also present. Due
to the large amount of data gathered, we do not provide information on the type of measurements for each snow depth dataset.

Following this initial compilation, we obtain 21,502 individual time series of snow depth, upon which we apply further
selection criteria before further use. The $\Delta$SNOW model (Winkler et al., 2021) requires a gap-free record of daily snow depth.
Therefore we use a robust gap-filling procedure to the HS time series (see Appendix A). We quality control the gap-filling based
on confidence criteria and we select stations with at least one entire gap-free snow year between 1950 and 2022. We define
a snow year from 1st of September to 31st of August, since this was the first day of the month with most snow free records
over the entire dataset. Furthermore, the $\Delta$SNOW model can only be reliably applied over seasonal snowpacks (Winkler et al.,
2021), and the SNOTEL dataset (used here for model regionalisation) contains only sites with a minimum of 40 days of
continuous snow cover. Because of the uncertainty in model performance for shorter snow cover durations, we retain only
stations where the mean continuous snow cover duration is at least 40 days. The number of stations in the dataset after quality
control and selection criteria is 11,071. An overview of the datasets and sources is available in Table 1 and the spatial coverage
is shown in Fig. 2b.

We are not allowed to republish the original HS data used as source to derive the SWE dataset because it is already freely
available (See references in Table 1). More information on the data sources and how to download the HS data can be found in
the data availability section.





## 3 Method: Snow depth to water equivalent conversion

### 3.1 Definitions

We show in Fig. 3 the terms to describe the snow season. The day count of a snow year starts on 1st of September, which is day 1. The annual snow season is the longest period of continuous snow cover (SWE > 0 mm) in a snow year. The maximum snow water equivalent value in the annual snow season is the peak SWE. The value of peak SWE can last for days while there is no more snow accumulation, until the snowpack starts to melt. Snowmelt is defined as a decrease in SWE. The timing of snowmelt onset corresponds to the last day of peak SWE and dividies the annual snow season into the accumulation season and the snowmelt season.

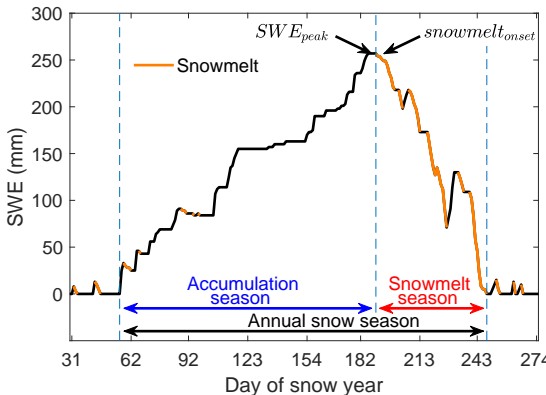

**Figure 3.** Snow season terms and definitions. $SWE_{peak}$ is the highest snow water equivalent value. Snowmelt onset is the last day of peak SWE and divides the snow season into accumulation season and snowmelt season.

### 3.2 Model choice and brief description

We use the $\Delta$SNOW model of Winkler et al. (2021) to convert snow depth (HS) to snow water equivalent (SWE) time series. We use the $\Delta$SNOW model because of its low complexity and little input data required, given that most snow depth time series do not contain other meteorological data (see Section 2). Furthermore, the $\Delta$SNOW model shows a high accuracy compared to other conversion models (Winkler et al., 2021), especially because of its ability to integrate the temporal evolution of snow depth into the model. Here, we further test its ability to accurately model the temporal dynamics of SWE by evaluating the model performance on other variables (see Section 3.6) than simulated daily SWE and simulated peak SWE, which were the main evaluated variables in previous studies (Hill et al., 2019; Winkler et al., 2021). In addition, the low number of required parameters makes it possible to regionalise the model and to transfer it to different types of snowpacks and climates, as shown in Section 3.4.

The model estimates total snowpack SWE by accumulating, compacting and drenching a series of snow layers, based on the temporal evolution of snow depth and 7 parameters that need to be calibrated. The model has four modules for HS computation,



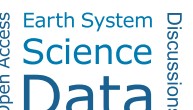

which are activated according to the value of $\Delta\mathrm{HS}(t)$ (the change in depth of the entire snow pack between $t-1$ and $t$), the density of the snow layers ($\rho_l$, which can reach a maximum density of $\rho_{\max}$), and a *threshold deviation* parameter ($\tau$). The four modules are:

(i) New snow and overburden module, activated if $\Delta\mathrm{HS}(t) > \tau$;

(ii) Dry compaction module, activated at every time step if $\rho_{lj} < \rho_{\max}$ for at least one snow layer $j$;

(iii) Drenching module if $\Delta\mathrm{HS}(t) < -\tau$ and runoff submodule when $\rho_l = \rho_{\max}$ for all snow layers;

(iv) Scaling module when $|\Delta\mathrm{HS}(t)| < |\tau|$.

Each increase in snow depth activates the new snow module and creates a new layer in the snowpack. The parameter *new snow density* ($\rho_0$) determines the density of the new layer. In case the snowpack already contains one or more layers, the overburden submodule is activated, which increases the density in underlying snow layers due to the weight of the new

snow. The *overburden parameters* ($k_{ov}$, $c_{ov}$) control this submodule. There is no maximum number of snow layers, and their thicknesses are determined by the temporal evolution of snow depth.

A snow accumulation event is generally followed by a decrease in snow depth due to snow metamorphism. This process densifies the snowpack but does not imply a decrease of SWE. This densification of snow layers is computed by the dry compaction module and occurs at every time step until all layers in the snowpack reach the *maximum snow density* ($\rho_{\max}$). The

densification is controlled by the *viscosity parameters* ($\eta_0$, $k$).

A decrease in snow depth also activates the drenching module, which simulates wet metamorphism and snowpack water percolation from the top to the bottom layers, further densifying the snowpack layers until all layers have reached $\rho_{\max}$. During dry compaction and drenching, SWE does not decrease even though snow depth does. When all layers have reached the *maximum snow density*, the runoff submodule is activated and a decrease in snow depth leads to a decrease in SWE (i.e.

snowmelt).

The scaling module intervenes when $|\Delta\mathrm{HS}(t)| < |\tau|$ and avoids excessive mass loss or mass gain from uncertainties in manual and automatic snow depth measurements. When $|\Delta\mathrm{HS}(t)| < |\tau|$, the model reevaluates dry compaction and makes small adjustments to HS within *threshold deviation $\tau$*. For further details on the model we refer the reader to Winkler et al. (2021).

## 3.3 ΔSNOW model calibration

The ΔSNOW model (Winkler et al., 2021) contains seven parameters that might be site-specific, or depend on local conditions. The model was originally developed and calibrated over 15 alpine sites in the Swiss and Austrian Alps with continuous measurements of snow depth (HS) and biweekly measurements of SWE. One unique set of optimized parameters was obtained by Winkler et al. (2021) (Table 2). Here we use continuous time series of daily HS and SWE from 812 sites from the SNOTEL

dataset (see Section 2.1) to calibrate the model, and we obtain one set of optimized parameters per site.





The *new snow density* ($\rho_0$) is the most sensitive model parameter, largely controlling the bias in daily SWE and peak SWE (Winkler et al., 2021). Our initial tests showed that the *maximum snow density* ($\rho_{\max}$) has a stronger impact on the timing of modelled peak SWE than on the modelled daily SWE, a sensitivity which was not explicitly tested in the work of Winkler et al. (2021). This sensitivity is not surprising because the *maximum snow density* determines the maximum densification of the snowpack, after which any decrease in snow depth will lead to a decrease in SWE. However, the magnitude of modelled peak SWE is still mostly controlled by the *new snow density* parameter. Furthermore, $\rho_0$ and $\rho_{\max}$ are the two parameters that exhibit some relationship with climate variables (see Section 3.4). Therefore, we focus here on the calibration of these two key parameters, $\rho_0$ and $\rho_{\max}$, and retain the values determined in the analysis of Winkler et al. (2021) (see Table 2) for the remaining five ($\eta_0$, $k$, $c_{ov}$, $k_{ov}$, $\tau$).

To identify the optimum values for $\rho_0$ and $\rho_{\max}$, we search within the original calibration ranges provided by Winkler et al. (2021), which are $\rho_0 \in [50, 200]\ kg\ m^{-3}$ and $\rho_{\max} \in [300, 600]\ kg\ m^{-3}$. We use latin hypercube sampling (McKay et al., 1979) to obtain 1000 parameter sets with $\rho_0$ and $\rho_{\max}$.

The performance of each model run is evaluated using the root mean square error of: daily SWE ($R_{\mathrm{daily}}$), peak SWE ($R_{\mathrm{peak}}$) and day of the year of peak SWE ($R_{\mathrm{peakdoy}}$). See Section 3.1 for the definition of peak SWE and day of peak SWE. We normalise each metric by the mean of the 1000 simulations to sum the three metrics into one single value to be minimised ($R_{\min}$). We also test different weighed sums of $R_{\mathrm{daily}}$, $R_{\mathrm{peak}}$ and $R_{\mathrm{peakdoy}}$ to minimise also the bias for the same three quantities. These tests lead to the following final objective function to minimise:

$$R_{\min} = 0.13 \cdot R_{\mathrm{daily}} + 0.45 \cdot R_{\mathrm{peak}} + 0.43 \cdot R_{\mathrm{peakdoy}} \tag{1}$$

Which ensures peak SWE and day of peak SWE have the largest weight on the overall model performance. This is because the model is unlikely to reach an optimal fit for peak SWE and day of peak SWE without a correspondingly good fit for daily SWE during the snow accumulation season.

For each of the 812 SNOTEL sites, the parameter set with the minimum $R_{\min}$ according to Equation 1 is retained.

### 3.4 Regionalisation of ΔSNOW model parameters

In order to make the modelled SWE estimates as transferable as possible to different types of snowpacks and climates where calibration is not possible, we used regional climate variables to estimate key model parameters. For each of the 812 SNOTEL sites, we compared the optimised parameter sets for $\rho_0$ and $\rho_{\max}$ against selected monthly climate variables from the World-Clim2 global dataset (Fick and Hijmans, 2017) which was used because of its high spatial resolution (~1 km) and global extent. Appendix B provides details on all the variables tested.

We further complement this analysis with the dimensionless climate index parameters proposed by Woods (2009) to analytically describe snow cover dynamics. These index parameters namely include $\overline{T}^*$, which is defined by Woods (2009) as the ratio between mean annual temperature and the amplitude of the annual temperature cycle.

We then use a stepwise linear regression to determine which variables best account for the variation in the ΔSNOW optimized model parameters $\rho_0$ and $\rho_{\max}$. This is similar to the approach used by Hill et al. (2019) to find explanatory climate


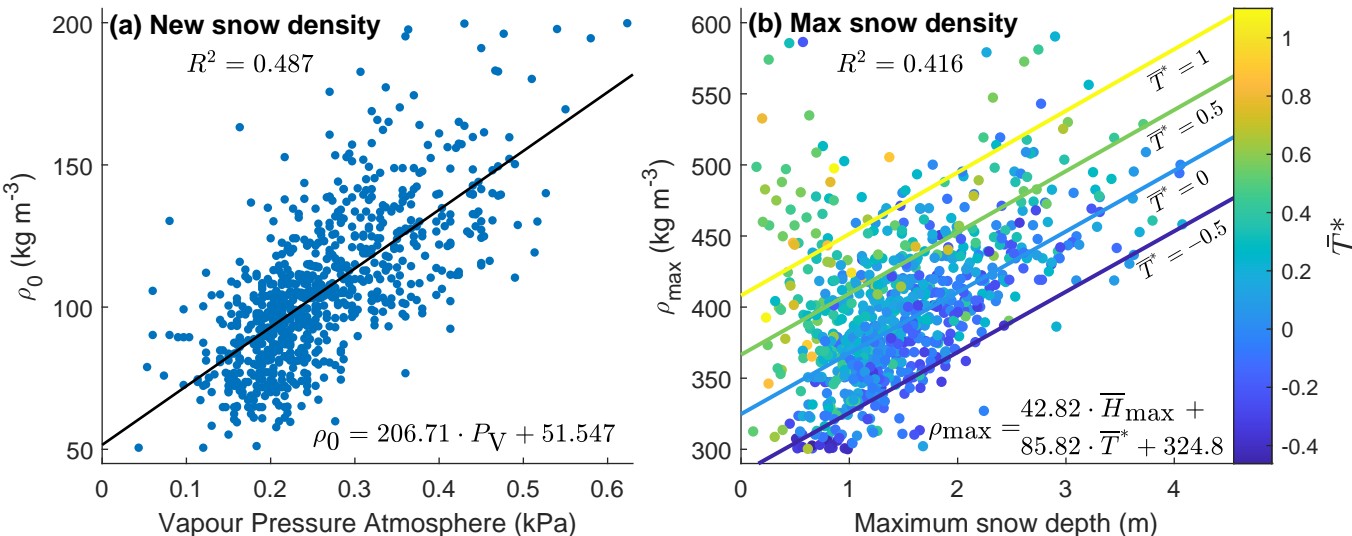

**Figure 4.** Regionalisation of $\Delta$SNOW model parameters. (a) fresh snow density ($\rho_0$) against mean December-January-February atmospheric Vapour Pressure ($P_V$) from WorldClim2 dataset. (b) maximum snow density ($\rho_{max}$) against maximum snow depth ($\overline{H}_{max}$) and dimensionless temperature index ($\overline{T}^*$). Equation 2 is shown in (a) together with the simple linear regression line. Equation 3 is shown in (b) together with regression lines for four values of $\overline{T}^* = [-0.5, 0, 0.5, 1]$. Adjusted r-squared displayed in each panel.

variables for the snow density parameters in their SWE estimation model. The stepwise linear regression finds the best ex-

planatory variable and includes additional variables if the adjusted $R^2$ improves by 0.02 or greater compared to its exclusion.

The final regression model for $\rho_0$ reads as (see also Fig. 4a):

$$\rho_0 = 206.71 \cdot P_V + 51.547, \tag{2}$$

The final regression model for $\rho_{max}$ reads as (see also Fig. 4b):

$$\rho_{max} = 42.82 \cdot \overline{H}_{max} + 85.82 \cdot \overline{T}^* + 324.76, \tag{3}$$

where $P_V$ is the mean December-January-February water vapour pressure in the atmosphere, which was taken from the WorldClim2 dataset (Fick and Hijmans, 2017), $\overline{H}_{max}$ is the average maximum snow depth in meters; and $\overline{T}^*$ is a dimensionless climate index defined by Woods (2009) and calculated based on WorldClim2 (Fick and Hijmans, 2017). See Appendix B for further description of the climate variables. We set a minimum of 54 and a maximum of 197 $kg\ m^{-3}$ for $\rho_0$ and a minimum of 309 and maximum of 580 $kg\ m^{-3}$ for $\rho_{max}$, as these were the minimum and maximum calibrated values (See Section 3.3).

The new snow density parameter $\rho_0$ is best explained ($R^2 = 0.49$) by the mean November-December-January (DJF) atmospheric vapour pressure. This influence is consistent with observed climatic controls for snow density, where more humid climates and maritime climates will typically have higher snow densities than drier climates, such as tundra and taiga (Bormann et al., 2013). The maximum snow density parameter $\rho_{max}$ is best explained ($R^2 = 0.42$) by a combination of the the average




**Table 2.** The ΔSNOW model parameters as originally calibrated by Winkler et al. (2021) and with our regionalised approach (Section 3.4).

| Model version | $\rho_0$ $(kg\ m^{-3})$ | $\rho_{max}$ $(kg\ m^{-3})$ | $\eta_0$ $(Pa\ s)$ | $k$ $(m^3kg^{-1})$ | $k_{ov}$ $(-)$ | $c_{ov}$ $(Pa^{-1})$ | $\tau$ $(cm)$ |
|---|---|---|---|---|---|---|---|
| $\Delta$SNOW$_{Original}$ | 81 | 401 | $8.5 \cdot 10^6$ | 0.030 | 5.1 | $0.38 \cdot 10^{-4}$ | 2.4 |
| $\Delta$SNOW$_{Regionalised}$ | Equation 2 | Equation 3 | | | | | |

annual maximum snow depth and $\overline{T}^*$. This conforms to broad expectations from snowpack dynamics, whereby the greater
mass of a deeper snowpack contributes to higher densification. An estimated value for average annual maximum snow depth
should always be available since the model needs continuous time series of snow depth. The dependence on $\overline{T}^*$ (ratio between
mean and amplitude of annual temperature) reveals an additional link between $\rho_{max}$ and the regional climate, whereby close to
zero $\overline{T}^*$ arises for high temperature amplitudes (typically continental climates) or for marginal snow areas with mean annual
temperatures close to zero. In contrast, high positive or negative $\overline{T}^*$ values correspond to low temperature amplitudes (typically
maritime climates) or highly negative or positive mean annual temperatures. This dynamic can be seen in Fig. 4, in the case of
two sites with equal maximum snow depth, the one in the warmer climate will have a higher *maximum snow density*, which
agrees with the literature (Bormann et al., 2013).

Based on the two linear regression models, we estimate the *new snow density* and *maximum snow density* parameters for all
the sites in the regionalisation, evaluation, and model input datasets (see Section 2) and run the ΔSNOW model for all of the
snow depth time series. The performance of the model using these regionalised parameters (ΔSNOW$_{Regionalised}$) is analysed in
Section 4.

The final modelled NH-SWE dataset presented in Section 5 is based on the parameters estimated using this approach.
Furthermore, we provide code for the user to extract the ΔSNOW$_{Regionalised}$ model parameters for any point in the Northern
Hemisphere. Latitude-longitude coordinates and a value for maximum snow depth need to be provided by the user (see code
and data availability section).

### 3.5  Comparison with an alternative SWE estimation approach

In order to provide broader context for our Northern Hemisphere SWE estimates, we compare these results with a previously
published statistical SWE estimation model (Hill et al., 2019) (from now on Hill model). The Hill model also estimates SWE
using snow depth data and climatological variables, but does not need a complete time series of snow depth and can instead
estimate single values based on an estimate of snow density. There is some similarity in the climatological variables found by
Hill et al. (2019) that best estimate snow density variation and those used in the present paper (see Section 3.4). The Hill model
estimates higher snow density for locations with a higher mean winter precipitation (humid climates), and a lower snow density
for locations with a high temperature difference between warmest and coldest month (continental climates). In addition, in the
Hill model, the day of the hydrological year contributes to estimating snow density, with higher densities towards the end of
the snow season, when the snowpack is deeper and more compact.





We use the WorldClim2 global dataset of monthly climate variables (Fick and Hijmans, 2017) to obtain the Hill model parameters, namely mean winter precipitation (PPTWT) and the temperature difference between the warmest and coldest month (TD). The reader is referred to the work of Hill et al. (2019) for more details on the model. We run the Hill model for all the sites in the regionalisation and evaluation datasets (see Section 2). The performance of the Hill model is analysed in
Section 4 as comparison along with the regionalised ΔSNOW model (ΔSNOW$_{\text{Regionalised}}$).

## 3.6   Extended model performance assessment

Previous snow depth to SWE conversion models have evaluated model performance based on root-mean-square-error and bias of daily SWE and of peak SWE, which we also employ to enable comparisons. This is mostly due to a lack of data to evaluate more metrics. For example, continuous measurements of SWE are required to assess the model performance on timing of
snowmelt onset, but these are rarely available. In order to assist with potential applications in water resources management and modelling, here we use our extensive HS-SWE data compilation (see Section 2) to assess additional aspects of the SWE model performance, namely: the relative bias of peak SWE, the daily SWE time-series variability, the timing of snowmelt onset, the total snowmelt and the snowmelt duration.

**Table 3.** Summary of model performance assessment variables and metrics. All performance variables are computed for each annual snow season. Snow seasons with non-daily SWE observations or with gaps do not allow the computation of snowmelt onset and snowmelt total. All the biases are computed as modelled minus observed.

| Performance variable | Performance metric | Description |
|---|---|---|
| SWE daily | RMSE (mm), Bias (mm) | The root-mean-square-error of all modelled daily SWE values. The mean difference (bias) between all modelled and observed daily SWE values. |
| SWE peak | RMSE (mm), Bias (mm), Bias (%) | The root-mean-square-error of modelled peak SWE values. The difference (bias) between modelled and observed peak SWE (see Fig. 3), also in percentage. |
| SWE time-series | NSE (-) | The Nash-Sutcliffe Efficiency (Nash and Sutcliffe, 1970) of daily SWE time series (see Eq. 4). One NSE value for each modelled annual snow season. |
| Snowmelt onset | Bias (days) | The difference (bias) between modelled and observed timing of onset of the snowmelt season (see Fig. 3). |
| Snowmelt total | Bias (mm) | The difference (bias) between modelled and observed cumulative total snowmelt for the annual snow season and for the melt season (see Fig. 3). |
| Snowmelt duration | Bias (days) | The difference (bias) between modelled and observed number of snowmelt days during the annual snow season and during the melt season (see Fig. 3). |

The performance variables and metrics together with their description are shown in Table 3. The reproduction of the daily
SWE time series dynamics is assessed based on the well-known Nash-Sutcliffe criterion (Nash and Sutcliffe, 1970), widely



used in streamflow model assessment:

$$R_{\text{NASH}} = 1 - \frac{\sum_{t=1}^{N}(x_{obs}(t) - x_{mod}(t))^2}{\sum_{t=1}^{N}(x_{obs}(t) - \overline{x_{obs}})^2}, \tag{4}$$

where $x_{obs}$ is the observed quantity (here daily SWE) at time step $t$, $x_{mod}$ is the corresponding modelled quantity and $\overline{x_{obs}}$ is the mean of all $N$ observed values.

## 4  Model performance and evaluation

Here we give a summary of the $\Delta$SNOW regionalised (see Section 3.6) model performance using both the regionalisation and evaluation datasets.

### 4.1  Daily SWE, peak SWE and snowmelt onset

The daily SWE values (Fig. 5a,b) and the peak SWE values (Fig. 5c,d) reproduce the observed values well and show a comparable performance to previous efforts to convert snow depth to snow water equivalent (Jonas et al., 2009; Sturm et al., 2010; Hill et al., 2019; Winkler et al., 2021). The regionalisation dataset is generally unbiased for daily SWE and peak SWE, while the evaluation datasets show a slight negative bias. This negative bias is largely driven by the Canadian SWE dataset because it contains a large amount of data and very deep snowpacks, but the median relative bias across the entire dataset is only $-11.3\%$ (Table 4). The mean relative bias across all datasets given equal weight is $-1.7\%$, and the largest relative bias is $17\%$ (Table 4). For the GCOS-CH and Kuhtai-AT datasets, the regionalisation of parameters yields a lower model performance for daily SWE and peak SWE than using the original $\Delta$SNOW model parameters (see Table C1). This is because these two datasets were used by Winkler et al. (2021) to calibrate the model. The timing of snowmelt onset is well reproduced by the $\Delta$SNOW$_{\text{Regio}}$ model, with a 1.45 day bias for the regionalisation dataset and a 1.34 day bias for the evaluation datasets (Fig. 5e,f). The positive bias in the timing snowmelt onset may indicate a very small model delay in the densification of all the snowpack layers to $\rho_{max}$. Nonetheless, the model performance is excellent considering no information on daily temperature or any other variable is used to model the start of the snowmelt season. For comparison, the regression model from Hill et al. (2019) has a negative bias of 15 days for the SNOTEL dataset and 17 days for the CanSWE dataset (see Table C1). This very premature onset of the snowmelt season occurs in the regression model because the day of peak SWE corresponds to the day of maximum snow depth, which is not realistic as snowpack compaction and ripening can still occur prior to melt loss. A period of snow depth decline but no change in SWE can last for many days. This can be observed in Fig. 6d. Thus, the improved capacity to capture accurate timing is a key advantage of using the $\Delta$SNOW model.

### 4.2  Daily SWE time-series

In terms of daily SWE dynamics, Fig. 6a shows that the $\Delta$SNOW$_{\text{Regio}}$ model can reach very high Nash-Sutcliffe Efficiencies as the bias in daily SWE approaches zero, all data considered. This is shown by the high density of points close to a NSE of 1 in Fig. 6a. In contrast, the regression model from Hill et al. (2019) shows lower NSE values in general for an unbiased

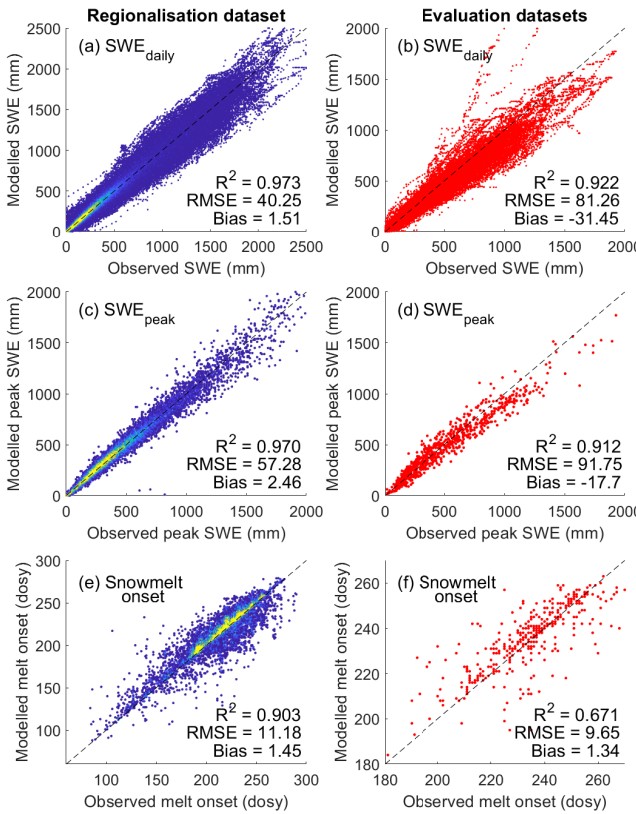

**Figure 5.** Model performance of $\Delta$SNOW$_{Regio}$. For the regionalisation (left column) and evaluation (right column) datasets: performance of all modelled daily SWE (a,b), modelled peak SWE (c,d) and day of snowmelt onset (e,f). Colours in left column show scatter density. dosy: day of snow year starting on 1st of September.

daily SWE performance (Fig. 6b), shown by a lower density of points close to a NSE of 1. Only the regionalisation dataset is analysed here, because many time-series in the evaluation datasets are not continuous daily measurements and thus do not allow NSE estimates for the entire snow season. The better performance of the $\Delta$SNOW$_{Regio}$ model is explained by the inclusion of snowpack metamorphism processes which are not included in regression models (e.g. compaction and ripening, see Section 3.2). An example can be seen in Fig. 6c and 6d, where an unbiased performance for both models shows a higher NSE (0.992) for the $\Delta$SNOW$_{Regio}$ compared to the regression model (0.960). The NSE of modelled daily SWE in Fig. 6c is close to 1, while the NSE of the regression model in Fig. 6d is lower and clearly shows that the model has difficulty accurately capturing snowpack changes due to daily snow accumulation and settling processes.

### 4.3 Snowmelt duration and total snowmelt

The general daily SWE dynamics are well reproduced by the $\Delta$SNOW$_{Regio}$ model. However, a negative bias in total snowmelt and snowmelt duration is observed when the full annual snow season is considered (Fig. 7a). This negative bias occurs because

**Table 4.** Performance across datasets for daily SWE, peak SWE and timing of snowmelt onset. For each dataset, the median performance of all stations is shown. SNOTEL is the regionalisation dataset, and the others are evaluation datasets (see Section 2).

| | SWE daily Bias (mm) | SWE peak Bias (mm), Bias (%) | Snowmelt onset Bias (days) |
|---|---|---|---|
| SNOTEL | 1.0 | 3.4 (−1.0%) | 1.6 |
| CanSWE | −34.0 | −56.4 (−11.3%) | 3.0 |
| GCOS-CH | −0.6 | 50.9 (17.2%) | - |
| RIHMI-WDC | −16.9 | −16.2 (−14.6%) | - |
| NVE | −15.3 | −32.5 (−5.9%) | −1.3 |
| Kuhtai-AT | 7.5 | 47.7 (12.7%) | −7.3 |
| Sodankyla-FI | −15.4 | −29.8 (−15.0%) | - |
| Col-de-Porte-FR | −28.8 | 9.7 (2.5%) | - |
| Mean | −12.8 | −2.9 (−1.7%) | −1.0 |

small snowmelt events (< 5 mm) during the accumulation season are not well captured by the $\Delta SNOW_{Regio}$ model. The model snow layers do not reach the maximum snow densification for a short period of snowmelt, and the 1 centimeter precision of snow depth measurements that drive the $\Delta SNOW$ model make the minimum modelled melt 3-5 mm (depending on the snow density). During the snowmelt season, the total snowmelt exhibits little to no bias, and the snowmelt duration has a slight negative bias. The negative bias in the duration of the snowmelt season is largely due to the positive bias in the timing of snowmelt onset (2.5 days on average), resulting in a shorter modelled total snowmelt season. Despite the slight negative biases shown for snowmelt duration and total snowmelt, the overall performance is very good and enables confidence in the application of modelled SWE using the $\Delta SNOW_{Regio}$ model. This confidence is not possible for regression model estimates of SWE, especially for cumulative variables such as snowmelt duration and total snowmelt, which can be greatly overestimated (although peak SWE magnitudes can be equally well estimated).



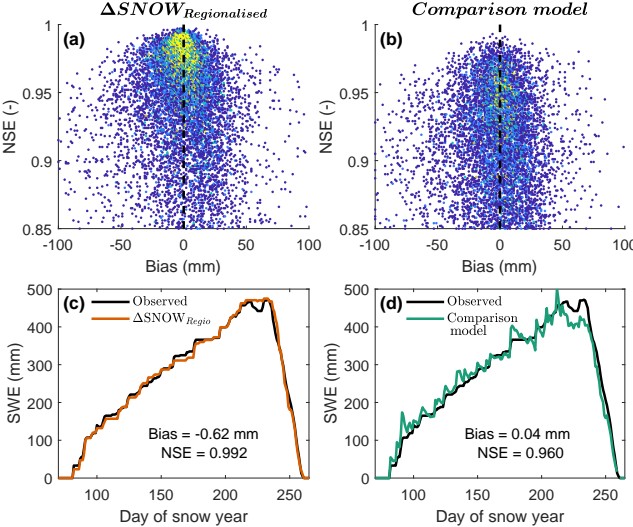

**Figure 6.** Model performance for daily time series. Nash and Sutcliffe Efficiency (NSE) vs Bias of daily SWE for (a) the $\Delta SNOW_{Regio}$ and (b) the comparison (Hill) model. Colours show scatter point density. One example time series of observed daily SWE and modelled daily SWE for (c) the $\Delta SNOW_{Regio}$ and (d) the regression model (Hill model) for comparison. Time series from the 2020 water year of SNOTEL station ID 371; Buck Flat, Utah, United States; Latitude: 39.1300; Longitude: −111.44; Elevation: 2,868 m.a.s.l.

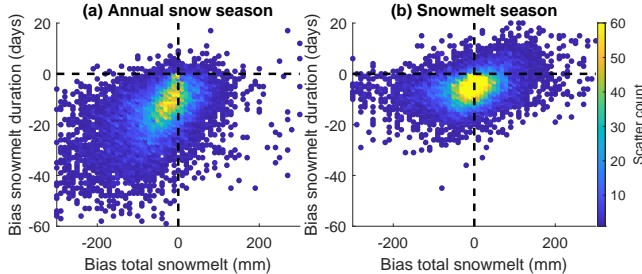

**Figure 7.** Bias in snowmelt duration vs bias in snowmelt depth for $\Delta SNOW_{Regio}$. (a) annual snow season bias and (b) snowmelt season bias only. See Section 3.6 for definitions of snowmelt season and biases.

## 4.4 Performance summary

The previous evaluation sections show model performance inclusive of all daily data from the regionalisation and evaluation datasets. In order to provide a more general overview, in Fig. 8 we display a summary comparing three model scenarios, regionalisation and evaluation datasets, and the six different performance metrics (see Table 3). The three model scenarios are the $\Delta SNOW$ model with its original parameters from Winkler et al. (2021), the $\Delta SNOW_{Regio}$ with our regionalised approach, and the Hill model (Hill et al., 2019). Mean performance values for each station are used for forming a boxplot in Fig. 8. This avoids over-weighting performance interpretations due to e.g. stations with very long time-series or outliers in model

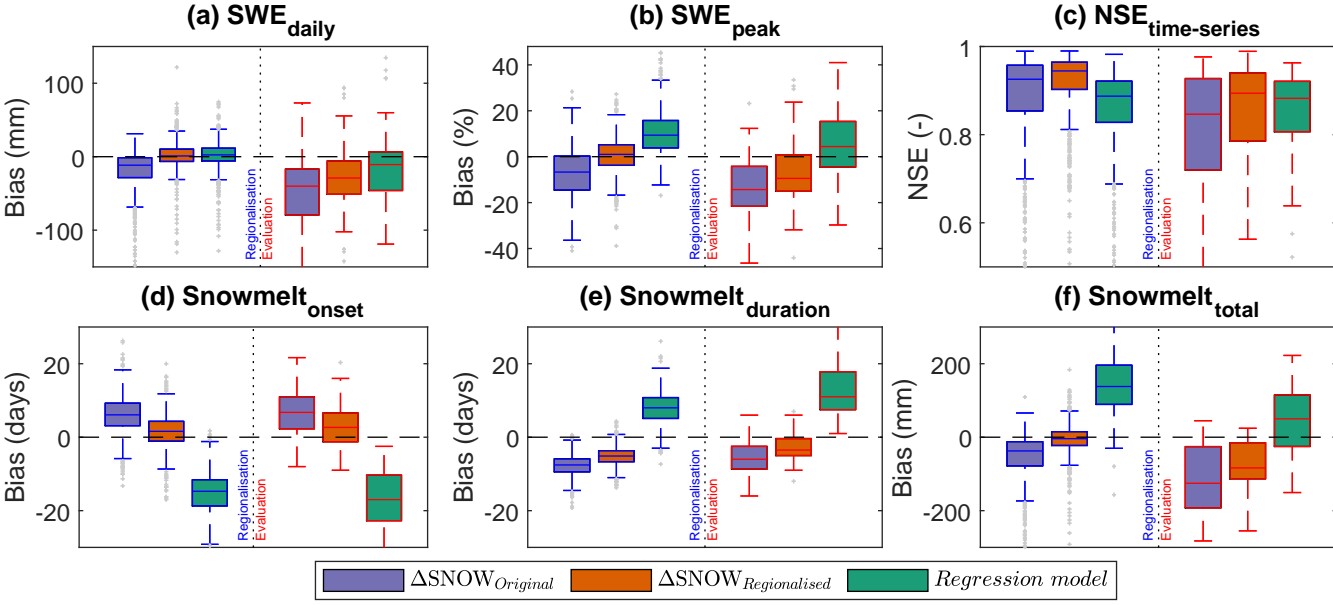

**Figure 8.** Model performance summary. Each subplot is split in two sides: the performance for the regionalisation dataset (left, blue lines) and the performance for the evaluation datasets (right, red lines). Boxes are built from mean values for all stations in each dataset, instead of considering all data points (i.e. a box for the regionalisation dataset is built from the mean performance for that model at each of the 812 SNOTEL sites).

performance. An important highlight from this comparison is the clear benefit in estimating $\rho_0$ and $\rho_0$ from regional climate
data ($\Delta\mathrm{SNOW}_{\mathrm{Regio}}$), which is apparent in the better performance across all 6 metrics compared to using the original values ($\Delta\mathrm{SNOW}_{\mathrm{Original}}$) obtained by Winkler et al. (2021). The full results for all these comparisons are also available in Table C1. The regionalised $\Delta\mathrm{SNOW}$ model performs better than the original $\Delta\mathrm{SNOW}$ model for 77% of all performance metrics and datasets, and better than the Hill model for 60% (see Table C1).

# 5  Final product: NH-SWE dataset

## 5.1  Snow climatologies and data characteristics

We present an overview of some of the key features of the final NH-SWE dataset in Fig. 9. The dataset contains 11,071 time series of daily SWE and estimated snow density at the point scale. The climatic range is broad, which can be seen in the large variability of peak SWE and of snow cover duration across the sites (Figure 9a,b). The deepest snowpacks (largest peak SWE) are generally located in mountainous areas, such as western North America, the European Alps, and Norway. Snowpacks are

generally shallower in relatively warmer locations where higher temperatures limit snowfall (e.g. southern parts of Canada, Germany, and Sweden). Shallow snowpacks are also found in very cold climates, where precipitation is limiting due to low





water vapour in the atmosphere (e.g. Siberia and northern Canada). Regarding snow cover duration, the spatial patterns are well linked to mean energy availability, with the higher latitudes and higher elevations having the longest snow cover duration.

In Figure 9c, we compare the links between peak SWE, snow cover duration, and $\overline{T}^*$. This demonstrates the large variation in

the interplay between precipitation and temperature in determining snowpack dynamics. All temperature regimes can produce a low peak-SWE if snow input (i.e. snowfall) is limited. For regimes with low peak-SWE, all snow cover durations are possible, depending on the temperature regime. In very cold climates (e.g. Siberia, Alaska), peak-SWE conditions can last for several months; in warmer climates (e.g. south Scandinavia), the snow will melt quickly.

On the contrary, there is a very clear link between snow cover duration and peak-SWE for temperature regimes that lead to

high peak-SWE. In fact, if peak SWE increases, snow cover duration also increase. Peak-SWE convergences to highest values towards intermediate $\overline{T}^*$, i.e. neither too cold nor too warm for large amounts of snow to occur and stay for long on the ground.

The median time series length in the dataset is 15 years, however the distribution is slightly bimodal with the overall average being 35 years, and a second peak in the availability of very long time series (up to 73 years) (Figure 9d). The distribution of site elevations reflects the large number of time series at low lying parts of coastal Norway, of Canada, and of Siberia.

However, more than 1,000 sites are located above 1,000 meters a.s.l., providing good representation of snow dynamics from higher elevations (Figure 9e).

## 5.2 Quality control and uncertainty

We provide quality control and uncertainty flags for the SWE time series to signal the values that should be handled with care. We flag the effect of the $\tau$ parameter (see Section 3.2) on the $\Delta$SNOW model when an increase in snow depth is smaller than

$\tau$ (2.4 cm). This means that sometimes there is a small increase in snow depth without an increase in SWE, leading to an artificial decrease in snow density. We correct this decrease in snow density and flag the value. Even though this is good for uncertain measurements, care should be taken for snow climates where small snow depth increments are normal. We also flag days when the computed snow density would fall below the fresh snow density parameter value. These are flagged as 1 in the quality control flag in the dataset.

We also flag all the values that result from gap-filling of snow depth. The gap-filling algorithm performs well overall (see Appendix A), however, we cannot guarantee that all gap-filled records are realistic. Special care is required if the data is filled across gaps longer than 15 days, because of the reduction in gap-filling performance (see Appendix A). We flag days that are gap-filled with the neural network algorithm with an "N", and we flag linearly interpolated gaps with an "L".

## 6 Usage and limitations

### 6.1 Potential applications

The SWE dataset presented here can be used freely and has a variety of potential applications. This data may be especially useful for applications where an accurate reproduction of the SWE dynamics in terms of timing of snow accumulation and melt

**(a) Mean Peak SWE**

**(b) Mean Continuous Snow Cover Duration**

**Figure 9.** Features of the NH-SWE dataset. (a) Mean yearly peak SWE at each station; (b) Mean continuous snow cover duration at each station; (c) Correlation between peak SWE, snow cover duration and $\overline{T}^*$. (d) Distribution of time series lengths in years; and (e) Distribution of station elevations.

is required. Additional uses include data assimilation into hydrological and climate models or validation of remote sensing products. A key strength of the dataset is accuracy in the timing of peak SWE and of snowmelt onset, which makes it valuable

for water resources assessment and management. The spatial extent of the dataset offers opportunities for snow climate research



and in particular to assess Norther-Hemispheric scale climate change impacts on snow water resources. We note, however, that some key snow regions are not yet included due to lack of long term data or a lack of data access (e.g. Himalaya, Central Asia, Tibetan Plateau).

The presented NH-SWE dataset is model-based, but based on actual measurements of snow depth, making this the only point
scale Northern Hemisphere SWE dataset based on in-situ snow data. Main advantages compared to other recently published Northern Hemisphere gridded SWE datasets (Luojus et al., 2021; Shao et al., 2022) are: i) NH-SWE includes mountains across the Northern Hemisphere, ii) it includes deep snowpacks exceeding 200 mm, iii) it has along temporal coverage, dating back to 1950s for some locations, iv) it uses actual point-scale observations of snow depth.

It is noteworthy that we do not only publish the NH-SWE dataset but also the underlying ΔSNOW model parameterization,
which can be used to estimate the model parameters for any point in the Northern Hemisphere. In addition, we provide code for this purpose along with the publication of the dataset on zenodo (see Code and Data availability), which enables any user to obtain the model parameters to run the ΔSNOW model of Winkler et al. (2021) for any given daily time series of snow depth.

## 6.2    Limitations

A few limitations should be considered before using the data or our regionalised parameterization of the ΔSNOW model of
Winkler et al. (2021). The ΔSNOW model requires good quality time series of snow depth (e.g. gap free, low measurement uncertainty). Measurement errors can accumulate and bias the SWE simulations for an entire season, especially if snow depth data is erroneous for a significant number of days. If the quality of the snow depth observations is poor or not continuous, we recommend using the regression model of Hill et al. (2019) to convert snow depth to SWE.

Our quality control flags capture gap-filled records and unrealistic snow densities, but the amount of data limits our capacity
of exhaustive quality checks; unrealistic time series could exist in the dataset. We show that our regional parameterization performs well especially in terms of snow cover dynamics and timings, but some SWE time series might nevertheless be slightly biased, resulting e.g. from an over or underestimation of model parameters. The user should also be cautious with any snow cover duration period of less than 40 days, or with snowpacks shallower than 50 mm of SWE. Even though we selected stations with an average of 40 days of snow cover, interannual variability could mean some years are shorter, making
the model performance uncertain (see Section 2.3). Snow sublimation and snow drift are also processes that are not captured by the ΔSNOW model.

Furthermore, the user should be aware of the spatial and temporal inhomogeneity of NH-SWE; the dataset covers most of the Northern Hemisphere, but some areas have a high station density (e.g. Scandinavia), while others show a rather low density (e.g. Siberia and Alaska). The time series lengths also vary from 1 to 73 years. Finally, NH-SWE does not include the original
snow depth data, which has to be directly downloaded from the data source, but we provide modelled snow density along with the SWE time series.



## 7 Conclusions

This paper presents NH-SWE, a dataset of Snow Water Equivalent (SWE) time series based on in-situ observations of snow depth that are freely available across the Northern Hemisphere. The dataset is based on an established model for continuous snow depth-to-SWE conversion (Winkler et al., 2021), for which we regionalized the model parameters for application at the Northern Hemisphere scale. The dataset contains 11,071 time series covering most areas of the Northern Hemisphere with a seasonal snow cover and spans the period 1950-2022.

The modelled daily SWE values and seasonal dynamics of SWE are generally well reproduced over the wide range of climates in the regionalisation and evaluation datasets (SNOTEL, CanSWE, GCOS-Switzerland, RIHMI-WDC, NVE, and other). The model has a slight delay towards a later start of snowmelt after the peak of snow accummulation, leading to a small negative bias in the duration of the melt season, but peak SWE and total snowmelt are generally unbiased.

Compared to SWE datasets based on remote sensing or climate reanalysis data, with a typical resolution of ∼10 km, NH-SWE is based on in-situ observations of snow depth on the ground and therefore provides a higher confidence in the magnitude and dynamics of snow accumulation and melt at the point scale.

This dataset can be used for a variety of applications that require reliable SWE estimates rather than snow depth, in particular in the fields of water resources and snow climate research, environmental and hydrological modelling, validation of remote sensing products or climate change impact research.

## 8 Code and data availability

The NH-SWE dataset is free to access and available at https://zenodo.org/record/7515603 (Fontrodona-Bach et al., 2023). The dataset is provided in two different formats (matrices and vectors) and contains modelled daily SWE, estimated snow density, and the quality control and gap-filling flags. An extensive metadata file includes information on station coordinates, elevation, length of time series, model parameters and the climate variables used to estimate them, and average snow climatology such as average maximum snow depth, average peak SWE and average maximum snow cover duration. The code used to estimate ΔSNOW regionalised model parameters can be found together with the NH-SWE dataset at https://zenodo.org/record/7515603 (Fontrodona-Bach et al., 2023).

**Appendix A: Snow depth data gap filling and quality control**

The ΔSNOW model (Winkler et al., 2021) requires a continuous record of daily observations of snow depth to estimate daily SWE. Our compilation of snow depth time series contains a large variability in time series length, number and size of gaps (Fig. A1). Many time series have missing data during the summer period when there is no snow (e.g. most MeteoSwiss stations).

This is easy to visually identify and manually fill with zeros, but it is complicated to fill all the series in an automated way with high confidence. Gaps also occur often in the middle of the snow season. Over 90% of the gaps in the dataset are less than 50 days long, and 80% of the gaps are less than 10 days long (Fig. A2).

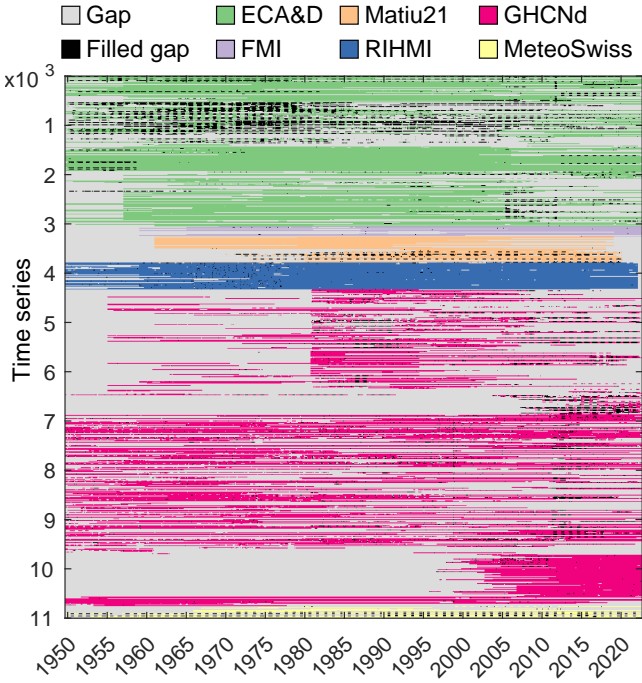

**Figure A1.** Final modelled NH-SWE dataset matrix. Each row corresponds to one of the 11,071 time series in the dataset at daily resolution. Colours indicate data sources, gaps, and filled gaps.

The high autocorrelation of daily snow depth time series (i.e. the correlation of snow depth at time $t$ to snow depth at time $t\text{-}i$) motivates us to use a *non linear autoregressive neural network with external input (NARX)* technique to fill the time series

of snow depth. Neural networks have been used by Silva et al. (2018) and Vieira et al. (2020) to fill environmental time series of sea surface wind speeds and ocean waves, but have to date not been applied to snow depth time series. We use the narxnet function from the MATLAB Deep Learning Toolbox (MATLAB R2021a). The NARX neural network uses snow depth at time steps up to $i$ days before time $t$ as input to estimate snow depth at time $t$. The delay parameter $i$ must be estimated. We found $i = 4$ days to be the best estimation of the delay parameter. This means that the snow depth at time $t$ is mostly influenced by

snow depth at 1, 2, 3 and 4 days before time $t$.

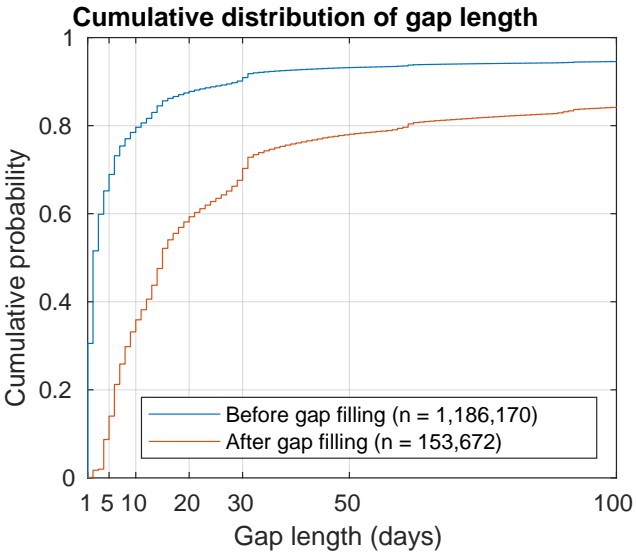

**Figure A2.** Cumulative distribution of dataset gap length before and after gap-filling. *n* is the total number of data gaps.

External inputs are used to help the NARX simulation. We tested a few NARX network architectures and inputs and found that including precipitation and temperature data as input to the NARX network showed promising results for filling the snow depth time series. We obtain daily precipitation and temperature data from DAYMET (Thornton et al., 2020) for the stations in North America and from E-OBS (Cornes et al., 2018) for the European stations. For the stations in Eurasia not covered by E-OBS we use temperature and precipitation station data from RIHMI-WDC (2022) stations. For stations where precipitation and temperature data cannot be obtained (i.e. locations not covered by gridded data and no station data available) we do not gap fill the snow depth data with NARX.

Neural network models require enough data to be trained. We train one model for each snow depth station in the dataset because the training of the network could differ per site. During training, the data is divided into a training, a validation, and a testing set. The division is made by blocks with a ratio of 60/20/20, meaning that for every 5 years, 3 are used for training, 1 for validation, and 1 for testing. For snow depth stations with more than 20 years of data where the trained model yields an r-squared performance higher than 0.9, we further test the performance of the NARX network gap filling technique by creating synthetic gaps of 1, 3, 7, 15, 30 and 50 days in the time series, at different times of the year. We then run the NARX network gap filling and compute the absolute error (in centimeters) of the filled snow depth compared to the real observed snow depth. We also compute the percentage error dividing by the observed snow depth value. For each site we repeat this procedure 50 times so that the gaps are randomly tested at different times of the year, for various years and for various gap lengths. The overall performance of the gap filling can be visualised in Fig. A3, where the mean absolute error and mean percentage error of the 50 simulations is plotted for each time of the year and gap length. At the beginning of the year (September-October) and at the end (June to August), the mean absolute error is zero, indicating that the NARX neural network performs very well



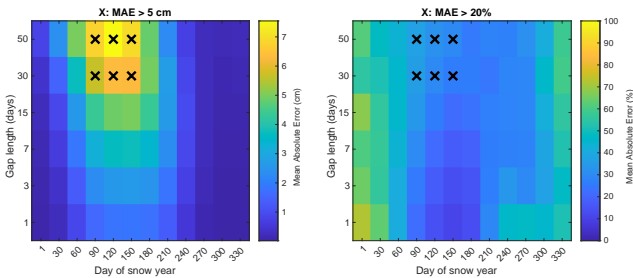

**Figure A3.** Mean performance of NARX gap-filling routine. Left panel shows absolute error in centimeters of the snow depth filled gaps at different times of the year and for different gap lengths. Right panel shows the same but in percentage error. A cross is displayed if the absolute error is higher than 5 cm and the percentage error higher than 20%.

during the summer periods when there is no snow. This allows filling the gaps in summer with high confidence. During the snow season (November to May), the errors are higher the longer the gap. In percentage terms, the error also becomes larger the longer the gap. We consider an acceptable error to be less than 5 cm or less than 20%. For gaps equal to or under 15 days the error of the NARX network gap filling is consistently under those thresholds. Figure A3 shows the mean over all sites where the performance is tested, but the procedure is applied to each site independently because the training of the network 475    could differ per site.

For each gap in each time series we decide whether or not to fill the gap based on a series of conditions. For stations with a minimum of 20 years of data where the performance is tested as explained above, we fill all the gaps under 16 days long because of the high confidence on filling them (Fig. A3). For gaps longer than 15 days and shorther than 70, we decide based on the performance according to the decision matrix (as in A3, where an acceptable error is less than 5 cm or less than 20%) 480    for that specific station. For long gaps between 70 and 365 days, we fill the gaps if the performance for gaps of 50 days is acceptable and at least 80% of the filled values are under 2 cm of snow depth, which means snow cover free periods are being filled. For stations with less than 20 but more than 5 years of data where the training of the NARX model yields an r-squared performance of at least 0.95, we also fill all gaps under 16 days long. For those stations, we also fill longer gaps if at least 80% of the filled values are under 2 cm, indicating that snow cover free summer is being filled. For stations where less than 5 years 485    of daily snow depth data are available but where the NARX model still yields an r-squared performance higher than 0.95, we fill all gaps shorter than 7 days and only fill gaps longer than 7 days if 90% of the filled values are under 2 cm.

After the NARX neural network gap filling, we also fill all gaps in the dataset that are equal to or shorter than 3 days by linear interpolation. If a station has more than 20 years of data but the NARX neural network model does not perform well, or no precipitation and temperature data are available, we compute the average main snow cover season and we fill with zeros 490    any gaps outside the average snow cover season. If after all the gap filling routine, one snow year at any station has only a gap of 7 days or shorter, we also fill the gap by linear interpolation. We do not fill gaps if any of the conditions above are not met.

Overall, 94.8% of the filled gaps are filled by the NARX neural network models, while only 5.2% are linearly interpolated or filled with zeros.





We quality control the time series of snow depth. We remove those years where the annual snow season (see Fig. 3) contains

more than 75% of filled gaps. We remove any snow cover period of less than 8 days that is fully gap-filled. We remove those years that are fully snow cover free, given that it is an unlikely event after our selection based on a minimum of 40 days of snow cover on average (see Section 2.3). We remove unrealistic daily snow depth increases of more than 150 cm in one day or decreases of more than 45 cm in one day. These values are based on the 99.99 percentile of all observed daily snow depth increases and snow depth decreases of the entire SNOTEL dataset (see Section 2.1). We remove years with perennial snow

cover, since the ΔSNOW model is developed over seasonal snowpacks. We remove all snow depth values that are higher than four times the median absolute deviation of the average yearly peak of snow depth.

Figure A4 shows example time series where the gap-filling is successfully filling the summer zeroes and small gaps during the snow season.







**Figure A4.** Example gap-filled snow depth time series. The title of each subplot corresponds to the station name, the country, and the elevation in meters above sea level.



## Appendix B: Climate variables for parameter regionalisation

We test a few climate variables to regionalise the $\Delta$SNOW model parameters in Section 3.4. All the climate variables are obtained from the WorldClim2 dataset of monthly climate variables (Fick and Hijmans, 2017). We complete the set of climate variables with snow climatology variables (i.e. climate variables based on snow depth data). The variables that were computed and tested are the following:

- $\overline{T}_Y$: Mean yearly temperature

- $\overline{T}_{DJF}$: Mean December-January-February temperature

- $\overline{T}_{NDJFM}$: Mean November-December-January-February-March temperature

- $\overline{P}_{DJF}$: Mean December-January-February cumulative precipitation

- $\overline{P}_{NDJFM}$: Mean November-December-January-February-March cumulative precipitation

- $\overline{P}_{snow}$: Precipitation as snow computed as the sum of monthly precipitation when temperature is below zero.

- $\overline{T}^*$: Dimensionless temperature index from Woods (2009) defined as the ratio between mean annual temperature and the amplitude of the annual temperature cycle.

- $\overline{P}^*$: Dimensionless precipitation index from Woods (2009) comparing the average precipitation rate and melt rate for a site.

- $P_{V,DJF}$: Mean December-January-February Vapour Pressure in the atmosphere.

- $P_{V,NDJFM}$: Mean December-January-February Vapour Pressure in the atmosphere.

- $S_{R,DJF}$: Mean December-January-February Shortwave Radiation.

- $P_{R,NDJFM}$: Mean December-January-February Shortwave Radiation.

- $\overline{H}$: Mean snow depth

- $\overline{H}_{max}$: Mean maximum snow depth (i.e. peak of snow accumulation)

## 525 Appendix C: Extended model performance and evaluation

This extends the model performance summary of Section 4, showing the mean model performance for each of the datasets separately, for three model scenarios ($\Delta$SNOW original, $\Delta$SNOW regionalised, and Hill model), and for the six evaluation metrics from Table 3.



**Table C1.** Extended model performance and evaluation. Here the performances are shown for all the datasets separately. $\Delta\mathrm{SNOW_{Orig}}$ refers to the original model parameters by Winkler et al. (2021), $\Delta\mathrm{SNOW_{Regio}}$ refers to our regional parameterization (see Section 3.4), Hill model refers to the regression model from Winkler et al. (2021). Where a value is not shown, it means that performance variable is not possible to compute for that dataset due to a lack of appropriate observed SWE records to compute that metric. The $\Delta\mathrm{SNOW_{Regio}}$ is better than $\Delta\mathrm{SNOW_{Orig}}$ on 77% of all datasets and metrics, and better than the Hill model on 61% of all datasets and metrics.

| | SWE$_{daily}$ | | | | | |
|---|---|---|---|---|---|---|
| | RMSE (mm) | | | Bias (mm) | | |
| | $\Delta\mathrm{SNOW_{Orig}}$ | $\Delta\mathrm{SNOW_{Regio}}$ | Hill model | $\Delta\mathrm{SNOW_{Orig}}$ | $\Delta\mathrm{SNOW_{Regio}}$ | Hill model |
| SNOTEL | 26.0 | 25.0 | 35.9 | −11.6 | 1.0 | 2.5 |
| CanSWE | 58.5 | 52.8 | 51.9 | −41.6 | −34.0 | −11.2 |
| GCOS-CH | 38.3 | 53.4 | 44.0 | −28.5 | −0.6 | −15.0 |
| RIHMI-WDC | 16.1 | 14.5 | 11.1 | −18.9 | −16.9 | −7.8 |
| NVE | 44.5 | 39.4 | 38.4 | −27.5 | −15.3 | 3.5 |
| Kuhtai-AT | 23.2 | 37.3 | 42.8 | −8.3 | 7.5 | 8.9 |
| Sodankyla-FI | 19.1 | 16.8 | 21.9 | −15.1 | −15.4 | 7.5 |
| Col-de-Porte-FR | 66.1 | 34.7 | 48.8 | −79.4 | −28.8 | −49.4 |
| Alptal-CH | 39.9 | 33.3 | 41.7 | −49.8 | −33.0 | −33.1 |

| | SWE$_{peak}$ | | | | | |
|---|---|---|---|---|---|---|
| | RMSE (mm) | | | Bias (mm), Rel. Bias (%) | | |
| | $\Delta\mathrm{SNOW_{Orig}}$ | $\Delta\mathrm{SNOW_{Regio}}$ | Hill model | $\Delta\mathrm{SNOW_{Orig}}$ | $\Delta\mathrm{SNOW_{Regio}}$ | Hill model |
| SNOTEL | 42.3 | 36.5 | 60.9 | −22.4 (−6.7%) | 3.4 (1.0%) | 33.3 (9.4%) |
| CanSWE | 86.3 | 73.0 | 59.4 | −68.6 (−16.1%) | −56.4 (−11.3%) | 3.9 (1.0%) |
| GCOS-CH | 56.0 | 76.9 | 89.7 | −3.0 (−0.6%) | 50.9 (17.2%) | 48.8 (15.4%) |
| RIHMI-WDC | 26.6 | 23.0 | 17.7 | −21.4 (−26.7%) | −16.2 (−14.6%) | 7.1 (13.7%) |
| NVE | 60.6 | 68.3 | 61.2 | −53.3 (−10.0%) | −32.5 (−5.9%) | 45.0 (13.1%) |
| Kuhtai-AT | 36.9 | 58.5 | 86.9 | 19.2 (5.1%) | 47.7 (12.7%) | 73.3 (19.5%) |
| Sodankyla-FI | 30.8 | 30.5 | 29.0 | −29.6 (−14.9%) | −29.8 (−15.0%) | 26.6 (13.4%) |
| Col-de-Porte-FR | 88.2 | 40.6 | 61.5 | −81.4 (−20.8%) | 9.7 (2.5%) | 30.8 (7.9%) |
| Alptal-CH | - | - | - | - | - | - |

| | Snowmelt onset | | | SWE time-series | | |
|---|---|---|---|---|---|---|
| | Bias (days) | | | NSE (-) | | |
| | $\Delta\mathrm{SNOW_{Orig}}$ | $\Delta\mathrm{SNOW_{Regio}}$ | Hill model | $\Delta\mathrm{SNOW_{Orig}}$ | $\Delta\mathrm{SNOW_{Regio}}$ | Hill model |
| SNOTEL | 6.1 | 1.6 | −14.7 | 0.93 | 0.94 | 0.89 |
| CanSWE | 6.0 | 3.0 | −17.1 | 0.85 | 0.91 | 0.88 |
| GCOS-CH | - | - | - | - | - | - |
| RIHMI-WDC | - | - | - | - | - | - |
| NVE | 9.1 | −1.3 | −12.9 | 0.81 | 0.84 | 0.85 |
| Kuhtai-AT | 2.2 | −7.3 | −22.5 | 0.94 | 0.89 | 0.84 |
| Sodankyla-FI | - | - | - | - | - | - |
| Col-de-Porte-FR | - | - | - | - | - | - |
| Alptal-CH | - | - | - | - | - | - |

| | Snowmelt total | | | Snowmelt duration | | |
|---|---|---|---|---|---|---|
| | Bias (mm) | | | Bias (days) | | |
| | $\Delta\mathrm{SNOW_{Orig}}$ | $\Delta\mathrm{SNOW_{Regio}}$ | Hill model | $\Delta\mathrm{SNOW_{Orig}}$ | $\Delta\mathrm{SNOW_{Regio}}$ | Hill model |
| SNOTEL | −37.4 | −4.0 | 138.0 | −7.6 | −5.1 | 8.0 |
| CanSWE | −125.4 | −83.6 | 62.8 | −5.5 | −4.0 | 11.5 |
| GCOS-CH | - | - | - | - | - | - |
| RIHMI-WDC | - | - | - | - | - | - |
| NVE | −116.2 | −63.5 | 31.8 | −8.5 | −2.5 | 10.0 |
| Kuhtai-AT | - | - | - | - | - | - |
| Sodankyla-FI | - | - | - | - | - | - |
| Col-de-Porte-FR | - | - | - | - | - | - |
| Alptal-CH | - | - | - | - | - | - |



*Author contributions.* AFB collected and processed the data, calibrated and parameterized the model, wrote the manuscript, produced all
the figures and results, and provided the SWE dataset. JL closely supervised the underlying PhD research and wrote the initial PhD project
proposal. JL and BS contributed to editing the text. AT had the initial idea of producing a SWE dataset based on depth data. BS, RW and AT
gave regular input on the work and the manuscript process. All co-authors have provided feedback on the manuscript.

*Competing interests.* The authors declare that they have no conflict of interest.

*Acknowledgements.* AFB acknowledges funding from the UK's Natural Environment Research Council (NERC) CENTA2 doctoral training
program. The computations described in this paper were performed using the University of Birmingham's BlueBEAR HPC service, which
provides a High Performance Computing service to the University's research community. See http://www.birmingham.ac.uk/bear for more
details.



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
