# Peer review of "NH-SWE: Northern Hemisphere Snow Water Equivalent dataset based on in-situ snow depth time series"

_Earth System Science Data, 2023_

## Author Response (AR1)

Dear Editors, Reviewers and Research Community,

We appreciate your time and valuable feedback on our manuscript. We have thoroughly revised the paper to address the comments and suggestions provided. Please find below a response to all points made by two reviewers and a community comment, including the actions taken on them. We also attach the corresponding author's tracked changes version of the manuscript.

We hereby submit a revised version of the manuscript, which we are confident has improved in quality and will meet your expectations for publication in ESSD.

Thank you again for your valuable input and suggestions.

Sincerely,

Adria Fontrodona-Bach and co-authors

(*Black and italics: reviewer comments.* Blue: Author's response. Blue underlined: Changes to manuscript)

Line numbers from the reviewer's comments refer to the original submitted manuscript, but line numbers from author's response refer to the **TRACK CHANGES** manuscript.

**RC1: Reply to Referee Comment 1 by Assoc. Prof. Chungyu Dong:**

*General comments:*

*Thank you for the opportunity to review the paper. This manuscript mainly presents a modeled SWE dataset based on in-situ snow depth observations for the Northern Hemisphere. Indeed, SWE is a critical hydrological variable while there are rare data for the snow-dominated areas. Hence, I think the newly generated SWE dataset is useful for the research community. The authors worked well in parameterizing the model and evaluating the data accuracy. It seems the data have a satisfactory quality for use. The methodology was described in detail. Overall, this is a good study and the paper was written well. In my opinion, the paper is publishable in the journal of ESSD. I provide some comments below, which may be helpful for the authors to further improve the paper.*

We highly appreciate the reviewer's time to review the paper and the positive report and suggestions to improve the manuscript. Find the responses to each comment and the actions taken below.

*Major points:*

*Fig. 2. Section 2. Technical comments: In my opinion, the grouping of the observation data is not quite reasonable, and I would suggest the authors further justify it. The authors only used the data from the SNOTEL stations in the western US for the model regionalization. They mainly applied the CanSWE station data from northwestern and northeastern Canada for model evaluation. This may make the model regionalization and evaluation easy technically. However, as the snow characteristics vary largely over different climate zones, vegetation bands, and terrains in the globe, it should be more reasonable to regionalize and evaluate the model using the data from more areas.*

We agree the grouping into regionalisation and evaluation of the model was simplified by using the SNOTEL dataset for regionalisation and the others for evaluation. The reason for this was to evaluate the model with fully independent datasets, rather than splitting the SNOTEL dataset in two groups. We thought this was the

best option because we later apply the regionalised model to a fully independent snow depth dataset to generate the NH-SWE. Furthermore, we were not able to find comparable data from other areas in the northern Hemisphere, considering the model evaluation requires continuous daily measurements of snow depth along with measurements (not necessarily continuous) of SWE or snow density. Although this provides some regional limitation, we are confident a broad spectrum of snow conditions are represented as the SNOTEL dataset contains a wide range of climates, as stated by Sun et al. (2019): "The 246 SNOTEL sites represent diverse hydroclimate conditions in the western United States: the Pacific Northwest's relatively mild and wet winter; the Intermountain West's continental climate; the Rocky Mountains' cold, snowy winter; California's Mediterranean climate; and the Southwest's arid climate." We also used the sites from higher latitudes in Alaska, further expanding the climatic range of sites.

Together with a similar remark by Reviewer 2 (RC2), we have emphasized and further justified our approach in the revised manuscript, lines 81-85, as well as lines 96-97. We have also expanded the description of the wide range of climates in the SNOTEL dataset (lines 91-93).

*Fig. 3. It seems the study assumed an ideal and single snow accumulation and melt pattern as shown in Fig. 3. In other areas, the snow processes might be more complicated than this one. Besides, even in the same location, the snow accumulation and melt may change greatly in different years. How would these changes affect the results?*

We agree with the reviewer that Figure 3 gives that impression. We have added a discussion about potential other types of snow seasons and how they can affect our definition of snowmelt season (lines 159-162). For the impact on the results, we have expanded Figure 6 by adding other snow seasons from the same location, so that the reader can see the effect of interannual variability on the model performance. We have slightly adapted the text in section 4.2 as the example time-series show now four full snow seasons.

*L191-194. The authors only calibrated the parameters of ρ0 and ρmax for the model. However, the other five parameters may also be sensitive to environmental and climate-type changes. Can you further evaluate the reliability of using fixed values of the five parameters of the entire Northern Hemisphere? Alternatively, please discuss the limitation and uncertainties.*

Indeed, our parameter calibration is restricted to ρ0 and ρmax. However, the other parameters did not show any clear sensitivity to the climate variables explored. Moreover, a comprehensive sensitivity analysis has already been completed by Winkler et al. 2021, which identified the high relative importance of the two density parameters compared to the others, which we have used as justification for the focus of our calibration efforts.

This reasoning was only briefly mentioned in the manuscript, so we have further clarified this (lines 212-214) to include discussion on the limitations and uncertainties of this approach (lines 446-452).

**Minor points:**

*The text in some figures and tables is too small and not clear, e.g., Fig. A3, Tab. C1*

We have expanded Fig. A3 to a two figure column. We have increased the size of Table C1 as much as possible to fit within one page.

**CC1: Reply to Community Comment 1 by Dr. Christoph Marty:**

*Thanks for the work. I liked reading the manuscript. I have two general comments and some specific suggestions.*

We appreciate the positive comments and suggestions from Dr. Christoph Marty to improve the manuscript. Please find our responses with actions taken below:

***General:***

*Two main uncertainties need to be added, probably in 6.2:*

- *The calibration is heavily dependent on the quality of the snotel data. It is long known and described in many studies (e.g. doi.org/10.1016/j.advwatres.2014.06.011 or Hill et al. (2019)) that daily snotel SWE can suffer from over-/under-measurement. Such errors are hard to detect and may also be responsible for processes described in L365.*

We acknowledge that SNOTEL is subject to errors as described in the literature. Avanzi et al. (2014) mainly describe issues with the hourly data, and Hill et al. (2019) also find biases and some errors in daily data. We used daily data and applied the same quality control as described in Hill et al. (2019) in order to guard against these known issues. The process described in our manuscript (L365 in preprint) shows that the ΔSNOW model considers uncertainty in the snow depth measurements when the daily snow depth change is smaller than 2.4 cm. However, some measurement errors will be larger and not detected.

We have added a paragraph discussing the uncertainty associated with SNOTEL data measurement errors in Section 6.2 (lines 427-432).

- *Some of the SWE data, especially those from the manual profiles (destructive method), may not have been taken at the exactly same spot as the HS data, which is usually read from a fixed installed stake, from a snow course or from an automatic snow depth measurements.*

We agree with the uncertainty associated with manual profiles that might not be taken at the exact same spot. We have specified the difference between automatic and manual measurements in section 2.2 (lines 115-119). We have also added a couple lines in 6.2 discussing the limitations of this (lines 433-435).

***Specific:***

- *Table 1: SWE_d for daily measurements.*

Corrected

- *Please, add either here or in 2.3 or in the reference, that snow data acquired from IDAWEB contains data from Meteoswiss and from the Institute of snow and avalanche research SLF (e.g. the station Kuehboden shown in Fig. A4 is an SLF station, please correct)*

We have added the SLF source acknowledgement in Table 1, in Section 2.3 (lines 126-127), and in the references.

- *L 183/4: It is important to prominently note that Δsnow_orginal parameters were obtained for the European Alps only. The authors confirm in the Conclusions, that "after calibration, the Δsnow model is widely usable"*

We have added that the original set of parameters was obtained for the European Alps only in lines 204 and 365.

- *L 240/41: I do not see data from two sites in Fig. 4?*

This was perhaps oddly phrased. We intended to help the reader interpreting Fig. 4, by hypothetically taking two of the many points in Fig 4b with a similar maximum snow depth (x-axis), and realising that the warmer climate (colour-axis) has a higher snow density (y-axis). We have rephrased it in lines 262-264.

- *L 291: …these two data sets were among others used…*

Corrected (line 317).

- *L 293: It should also be mentioned that according to Table C1 for daily SWE Δsnow_regio had a similar or slightly worse performance for other data sets (like CanSWE or Sodankyla) compared with Hill et al. (2019).*

Added (lines 318-319).

- *Table 4: SWE peak is not really measured by the bi-weekly measurements of the GCOS-SWE data set. So I don't know how fair the calculated bias measures are?*

The reviewer is right that we forgot to specify how we estimated peak SWE for the datasets with biweekly measurements of SWE such as the GCOS-SWE dataset. We took the largest biweekly SWE measurement in the snow season and compared it with modelled SWE on that same date. This is explained now in line 149.

The calculated biases are therefore for an estimated peak SWE, and provide valid bias measures in terms of SWE magnitude estimates, but not in terms of timing, which we do not consider for these bi-weekly measurements (hence the gap in Table 4 for snowmelt onset for GCOS-CH). In addition, the biases found for those datasets are similar to the ones found in the datasets with daily SWE measurements. A discussion about this has been added in Section 6.2 (lines 436-439).

- *L 336: Again Δsnow_orginal was obtained for the European Alps!*

Corrected.

- *Figure 9: I'd suggest explaining in the text as illustrative example why there two dark blue dots in the middle of the yellow dots somewhere in DE or AT in Fig. 9a & 9b. I assume, these are 2 high elevation stations surrounded by low elevation stations.*

They are indeed higher elevation stations surrounded by lower elevation stations in Poland. It is now explained in lines 374-375.

- *L 455: Not gap filled, means also not use for modelling SWE?*

For stations that were not gap-filled, we only modelled SWE for those years with continuous daily measurements of snow depth. If there were none, the stations was not used for modelling SWE. Clarified in lines 508-509.

- *L 494: What is the difference to the previous paragraphs. They were also about snow depth?*

The previous paragraphs were on the gap-filling method, but we also apply a quality control of the gap-filling after that, as explained in the paragraph. This is made clearer in line 546 now.

**RC2: Referee Comment 2 by Anonymous Referee:**

*This study is a valuable addition to the existing dataset of snow water equivalent (SWE) over the Northern Hemisphere. Overall, the paper is well-structured and the approach is clearly explained. I suggest the paper be published after the authors address the provided comments..*

We appreciate the reviewer found the dataset and manuscript valuable, and we thank the reviewer for the comments to improve the paper. Below we address the comments and describe the actions taken:

1.  *There are some recent papers that are highly relevant to this work, but the authors did not include them.*
    o   *Sun et al. (2019), which describes the development of regionally coherent snow parameters for a mass and energy balance snow model over the Western U.S. SNOTEL stations. Importantly, this paper emphasizes the biases in the SNOTEL dataset, including undercatch of snowfall, warm bias, and others. To enhance the quality of this work, I recommend that the authors explore the potential of using a QAQC SNOTEL dataset. The dataset is available for download at: https://www.pnnl.gov/data-products*
    o   *Sun et al. (2022), which introduces the gridded SWE dataset over the Continental U.S produced by a physics-based snow model. This work also introduces the regionalization of snow parameters based on climate variables.*
    o   *Zeng et al. (2018), which describes a gridded (4-km) daily SWE data over the Continental U.S by assimilating in situ measurements of SWE from SNOTEL stations, snow depth from thousands of NSW COOP stations.*
    o   *Dawson et al. (2017), which describes an approach to converting snow depth to SWE.*

We agree with the reviewer that we had missed these relevant references. We have introduced them appropriately in the Introduction. Sun et al. (2019, 2022) and Dawson et al. (2017) in lines 56-58, and Zeng et al. (2018) in line 40.

Regarding the use of the Bias Correction and Quality Control SNOTEL data (BCQC SNOTEL), we have explored if this would potentially improve the calibration and regionalisation of the model. We agree that the BCQC SNOTEL dataset provides an improved quality with respect to the raw SNOTEL dataset. However, we had already applied a quality control to the SNOTEL dataset (same as in Hill et al. 2019, see Line 87-88 in our original manuscript), and we only used SNOTEL stations and years with a continuous record of daily snow depth. This control has already highly reduced the frequency of poorer quality data. Nevertheless, we have downloaded the BCQC SNOTEL dataset and we have compared it with the quality controlled daily SNOTEL data that we used. In the figure below left panel, the x-axis is the filtered SWE data used in our manuscript, and in the y-axis the SWE data from BCQC SNOTEL. Indeed, there are many points that differ a lot, but the scatter density in the 1:1 line is over 10 000, thus the overall variation is very minor and the difference between the two is smaller than 1 mm for over 99% of the points, as can be seen in the cdf plot in the right panel. We therefore believe that using BCQC SNOTEL would not change our results or our model regionalisation significantly given our quality control version is already remarkably similar.

Nevertheless and also in response to a similar remark by the Community Comment 1, we have added a discussion about uncertainty associated with SNOTEL data measurement errors in Section 6.2 (lines 427-432).

[Figure]

2. *Is there a specific reason why the authors only used the SNOTEL dataset for regionalization? Since the SNOTEL stations only represent Western U.S. mountain ranges, I believe incorporating evaluation data that represent diverse geography and climate regimes into the regionalization process would improve the transferability of the results across the Northern Hemisphere. I recommend including all HS-SWE data in the regionalization.*

The rationale for using the SNOTEL dataset only for regionalisation was to keep the datasets for model regionalisation and model evaluation independent, since this is ultimately what we did to generate the NH-SWE dataset (i.e., we regionalised the model with SNOTEL, evaluated it with 8 independent datasets (Table 1), and applied it to our independent collection of Northern Hemisphere snow depth datasets (Table 1)). If we used all the HS-SWE data to regionalise the model, we would potentially lose confidence in the application to an independent dataset, as we would no longer have a technically independent evaluation. In addition, and similar to the response to a comment from Reviewer 1, the SNOTEL dataset does contain a wide range of hydroclimates to allow generation of a regional parameter set that can be tested for broader applicability with the independent dataset. As stated by Sun et al. (2019): "The 246 SNOTEL sites represent diverse hydroclimate conditions in the western United States: the Pacific Northwest's relatively mild and wet winter; the Intermountain West's continental climate; the Rocky Mountains' cold, snowy winter; California's Mediterranean climate; and the Southwest's arid climate." We also used the sites from high latitudes in Alaska, further expanding the climatic range of sites.

Together with a similar remark by Reviewer 1 (RC1), we have emphasized and further justified our approach in the revised manuscript, lines 81-85, as well as lines 96-97. We have also expanded the description of the wide range of climates in the SNOTEL dataset (lines 91-93).

3. *The paper's evaluation lacks spatial context, despite the availability of extensive spatial data. The model performance evaluation figures (Figures 5-7) aggregate all data across sites and time periods, precluding bias evaluations between sites. To better understand the spatial variation in the SWE error measured by different metrics, I suggest the authors add figures that display each error metric for each location on a spatial map, similar to the maps in Figure 9. Furthermore, the authors should provide an interpretation of the results to enhance the reader's comprehension.*

We believe a full spatial evaluation of the model outputs is beyond the scope of this paper. Our data is spatially distributed, and we focus on the performance of our modelling approach across different datasets. A thorough spatial evaluation of the regionalisation of the ΔSNOW model parameters can be the focus of future research applications, even at smaller regional or catchment scales. Nevertheless, we have added maps of spatial distribution of model performance in Figure C1 in the Appendix C. We have also added some lines in the results (evaluation section) about the spatial distribution of model performance (lines 311, 313-314, 321-322).

4. *Please add the mean performance to Table 4.*

Sorry we realised the table caption says median while in reality we are providing the mean. It is corrected now.

*References:*

*Dawson, N., Broxton, P., & Zeng, X. (2017). A New Snow Density Parameterization for Land Data Initialization. Journal of Hydrometeorology, 18(1), 197–207. https://doi.org/10.1175/JHM-D-16-0166.1*

*Sun, N., Yan, H., Wigmosta, M. S., Leung, L. R., Skaggs, R., & Hou, Z. (2019). Regional Snow Parameters Estimation for Large-Domain Hydrological Applications in the Western United States. Journal of Geophysical Research: Atmospheres, 124(10), 5296–5313. https://doi.org/10.1029/2018JD030140*

*Sun, N., Yan, H., Wigmosta, M. S., Coleman, A. M., Leung, L. R., & Hou, Z. (2022). Datasets for characterizing extreme events relevant to hydrologic design over the conterminous United States. Scientific Data, 9(1), 154. https://doi.org/10.1038/s41597-022-01221-9*

*Zeng, X., Broxton, P., & Dawson, N. (2018). Snowpack Change From 1982 to 2016 Over Conterminous United States. Geophysical Research Letters, 45(23). https://doi.org/10.1029/2018GL079621*